# LIQT: Bridging Liquid Neural Dynamics and Human Perceptual Mechanisms for Blind Image Quality Assessment

## Abstract

Blind Image Quality Assessment (BIQA) seeks to predict perceptual quality in reference-free scenarios, yet conventional methods often hard to capture the human visual system's adaptive spatio-temporal integration of degradation patterns. Inspired by the adaptive temporal dynamics of biological neural circuits, we propose Liquid Image Quality Transformer (LIQT), a novel BIQA framework that integrates Liquid Neural Networks (LNNs) with Transformer-based architectures. LIQT incorporates Liquid Self-Attention (LSA) equipped with Closed-Form Continuous-Time Module (CFCTM), which reformulates liquid time-constant neurons into stable closed-form solutions through learnable decay rates and Padé approximation, thus enabling LIQT to dynamically modulates feature extraction based on local image features. To emulate multi-scale perceptual evaluation, a Multi-Scale Image Quality-Aware Decoder (MIQAD) aggregates multi-scale features from LIQT for comprehensive quality regression. This work pioneers the integration of biomimetic neural mechanisms into BIQA and experiments in six benchmark datasets that span various types of distortion and image content demonstrate the superior performance of LIQT over state-of-the-art methods.

## 1 Introduction

Image quality assessment (IQA) aims to evaluate perceptual quality in alignment with human judgment, serving as a critical tool for optimizing image processing algorithms and benchmarking visual content fidelity Wang et al. (2004). Based on the availability of the pristine reference image, IQA can be typically divided into full-reference IQA (FR-IQA), reduced-reference IQA (RR-IQA), and no-reference or blind IQA (BIQA) Moorthy & Bovik (2011). FR-IQA and RR-IQA rely on complete or partial reference images, limiting their applicability in real-world scenarios where pristine references are typically absent. BIQA has garnered increasing attention by addressing this limitation through its reference-free operation, yet it remains inherently challenged in modeling the nonlinear relationship between distortions and human perception Yang et al. (2019).

BIQA task exhibits unique characteristics distinct from conventional visual tasks, particularly in its manifestation of disjoint processing of spatio-temporally continuous degradation information, where temporal memory updating and spatial feature extraction operate in decoupled optimization spaces Zhang et al. (2023), we refer this challenge as **"Spatio-Temporal Representation Disentanglement (STRD)"**. The human visual system (HVS) accomplishes quality evaluation through continuous-time neural dynamics van den Branden Lambrecht (1996), as illustrated in Figure 1(a), the HVS integrates historical perceptual experiences via spatio-temporal memory consolidation mechanisms to generate adaptive assessments for varying degradation images of the same type of object. This adaptive capability is driven by HVS's neural sensitivity to spatial contextual correlations and temporal persistence representations Yan et al. (2020). However, current CNN-based or Transformer-based BIQA methods hard to establish continuous memory mapping across distortion types and degradation levels for BIQA tasks because of this STRD challenge.

Liquid Neural Networks (LNNs) implement a biomimetic framework inspired by the neurophysiological mechanisms of *Caenorhabditis elegans* Hasani et al. (2020), employing Liquid Time-

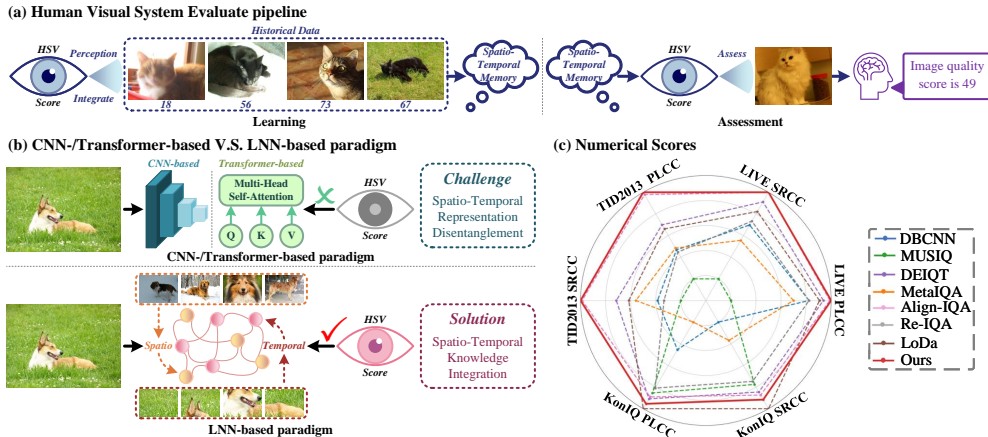

Figure 1: (a) Human Visual System Evaluate pipeline, the HVS integrates historical perceptual experiences via spatio-temporal memory consolidation mechanisms to generate adaptive assessments for varying degradation images of the same type of object. (b) The CNN-/Transformer-based paradigm suffer from STRD challenge, hard to establish continuous memory mapping across distortion types and degradation levels for BIQA tasks. LNN-based paradigm address the STRD challenge through Spatio-Temporal Knowledge Integration. (c) Comparison of seven state-of-the-art methods and LIQT performance based on the average values of PLCC and SRCC across three benchmark datasets.

Constant (LTC) neurons with input-dependent synaptic gating to dynamically modulate temporal dynamics through closed-form solutions Hasani et al. (2022). This approach has demonstrated efficacy in sequential decision-making and adaptive control tasks by emulating biological neural circuit behaviors Lechner et al. (2020); Chahine et al. (2023). Despite their demonstrated efficacy, LNNs exhibit a limitation in that they primarily focus on temporal dynamics and sequential processing, thus constraining their ability to effectively extract and interpret complex deep features and spatial contextual information within individual images.

To this end, we propose a novel BIQA method, Liquid Image Quality Transformer (LIQT), a framework that embeds LNNs into a Transformer-based architecture. Specifically, the Closed-Form Continuous-Time Module (CFCTM) reformulates liquid neuronal dynamics into stable closed-form solutions, replacing iterative differential equation solving with learnable decay rates and Padé approximations. CFCTM integrates into the Liquid Time-Constant Transformer (LTCFormer), each LTCFormer block uses Liquid Self-Attention (LSA) to combine CFCTM with self-attention, enabling adaptive feature processing based on local image features. Finally, a Multi-Scale Image Quality-Aware Decoder (MIQAD) aggregates features across hierarchical stages to emulate the simultaneous evaluation of fine details and global composition by human observers. Our LIQT model is designed with computational efficiency in mind, achieving strong performance with a significantly reduced parameter count.

In summary, the contributions of this paper are the following:

- We propose the Liquid Image Quality Transformer (LIQT), the first framework to incorporate LNNs into BIQA task, which integrates continuous-time neural dynamics through CFCTM, enabling adaptive temporal processing aligned with human visual mechanisms.

- We propose the Liquid Time-Constant Transformer (LTCFormer), extend the principles of adaptive temporal scaling in sequential tasks to spatial domains.

- Inspired by the LNNs, we introduce a novel closed-form implementation of liquid time-constant neurons, built on closed-form solutions from continuous-time network research, enabling stable integration with Transformer-based architecture.

- We verify our lightweight LIQT on 6 benchmark IQA datasets involving a wide range of image contents, distortion types, and dataset size. LIQT outperforms other competitors across all these datasets.

## 2 RELATED WORKS

**Liquid Neural Networks.** Recent advances in LNNs have demonstrated their potential to model dynamic systems through continuous-time differential equations and closed-form approximations. Hasani et al. (2020) introduced Liquid Time-Constant (LTC) networks, leverage input-dependent synaptic gating inspired by the neurophysiology of *Caenorhabditis elegans*, enabling adaptive temporal dynamics for sequential tasks such as autonomous navigation and prediction of time series. Lechner et al. (2020) demonstrated auditable autonomy in autonomous vehicles using compact, interpretable architectures with only 19 neurons, underscoring LNNs' efficiency and transparency. Hasani et al. (2022) further improved computational efficiency by replacing iterative differential equation solvers with analytical approximations, achieving speed improvements while retaining robustness and causal reasoning capabilities. Karn et al. (2024) have expanded LNNs applications beyond sequential tasks to non-causal domains, creating a unified mathematical framework that bridges temporal and spatial processing. Ayoub et al. (2024) have explored how the adaptive properties of LNNs can enhance learning in dynamic environments by leveraging input-dependent time constants to mitigate catastrophic forgetting.

**Blind Image Quality Assessment.** The development of deep learning has advanced the field of image quality assessment (IQA). Early IQA methods depended on handcrafted features for quality evaluation. However, this approach couldn't handle the complexity of blind image quality assessment (BIQA) tasks. Some CNN-based methods have achieved good results in BIQA tasks Saha et al. (2023); Zhao et al. (2023), but still struggle with CNNs' tendency to focus on local features, making it difficult to obtain an overall quality score for the image. Vision Transformers Dosovitskiy et al. (2021) have provided a new solution for BIQA, achieving good results through their excellent global context understanding Chen et al. (2024). Multi-scale feature adaption, cross-attention, or comparison technique have been used to solve the inherent efficiency issues in ViT Qin et al. (2023); Ke et al. (2021). Recently, state space models have emerged as an alternative approach, with QMamba Guan et al. (2025) demonstrating the effectiveness of selective state space mechanisms in capturing long-range dependencies for quality assessment while maintaining computational efficiency.

## 3 METHODOLOGY

### 3.1 OVERALL ARCHITECTURE

The overall architecture of the proposed Liquid Image Quality Transformer (LIQT) is illustrated in Figure 2, consists of three components: Liquid Time-Constant Transformer (LTCFormer), Closed-Form Continuous-Time Module (CFCTM), and Multi-Scale Image Quality-Aware Decoder (MIQAD). LTCFormer processes input images via window tokenization, splitting images into patch tokens and embedding them into spatiotemporal representations. CFCTM dynamically models temporal responses using LTC neurons, simulating biological visual processing through learnable decay rates and adaptive gating mechanisms. Each LTCFormer block uses Liquid Self-Attention (LSA) to combine CFCTM with self-attention, enabling adaptive feature processing based on local image features. The LTCFormer employs a four-stage architecture to extract multi-scale features, progressively generating representations with different dimensions. These hierarchical feature maps are subsequently processed by the MIQAD module, which conducts multi-scale quality assessment through global average pooling followed by scale-specific quality score prediction using MLP-based regression.

### 3.2 LIQUID TIME-CONSTANT TRANSFORMER

To maintain efficiency, former Transformer-based networks generally employ static attention mechanisms within small patches Chen et al. (2021); Dosovitskiy et al. (2021) or local windows Liu et al. (2021; 2022), potentially hindering the simulation of adaptive temporal dynamics inherent in visual processing. Motivated by the success of LNNs in continuous-time dynamic modeling Hasani et al. (2020); Lechner et al. (2020), we propose the Liquid Time-Constant Transformer (LTCFormer) to enhance visual models' continuous-time modeling capabilities.

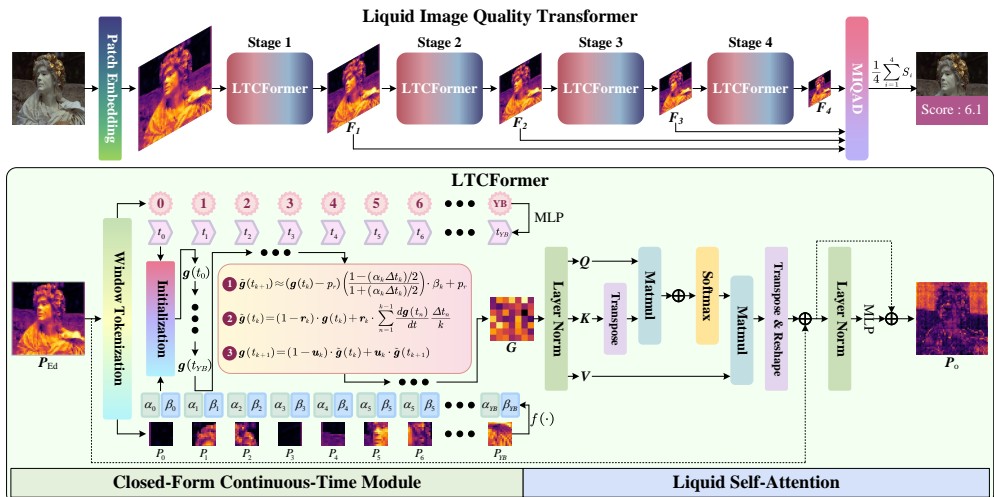

Figure 2: The overall framework of the proposed LIQT.

### 3.2.1 Window Tokenization

As illustrated in Figure 2, given an input image $\boldsymbol{I} \in \mathbb{R}^{B \times H \times W \times C}$, where $B$, $H$, $W$, and $C$ denote the batch size, height, width, and channel dimensions, respectively. We first perform the patch partition operation with the patch size of $d$ and flatten each patch into a set of patch tokens $\boldsymbol{P} \in \mathbb{R}^{B \times (HW/d^2) \times C}$. The HVS evaluates images through a neural process where both spatial context and temporal memory interact dynamically, rather than treating each region independently Chandler (2010). To model this integrated process in our framework, we transform spatial relationships into a temporal sequence by organizing image patches into windows where position encodes temporal ordering. This approach addresses the STRD challenge by mapping spatial adjacency relationships to sequential processing steps, allowing our model to leverage the continuous-time dynamics of LNNs to simulate how human observers progressively integrate local features with contextual information. Specifically, we partition patches into $Y = (H/dT) \times (W/dT)$ windows of size $T \times T$, where $Y$ refers to the number of windows. A linear embedding layer then projects these patches into a dimensional space $L$, generating patch embeddings $\boldsymbol{P}_{\text{Ed}} \in \mathbb{R}^{YB \times T^2 \times L}$. Subsequently, we perform the window partition operation on $\boldsymbol{P}_{\text{Ed}}$ to obtain the partitioned $\boldsymbol{P}_{\text{w}} \in \mathbb{R}^{YB \times T^2 \times L}$ and the position embedding $\boldsymbol{M} \in \mathbb{R}^{YB \times T^2 \times T^2}$ that preserves the relative position of a patch for the windows. The obtained $\boldsymbol{P}_{\text{w}}$ and $\boldsymbol{M}$ are then fed into CFCTM to model dynamic temporal responses in continuous time.

### 3.2.2 Closed-Form Continuous-Time Module

The human visual system (HVS) processes visual stimuli through complex neural interactions that evolve continuously over time, adapting to local image features Chandler (2010). In BIQA tasks, simulating this adaptive mechanism is crucial to improve the effect of quality evaluation. We first define the Liquid Time-Constant (LTC) neurons that form the basis of our approach, the membrane potential $\boldsymbol{g}(t)$ of LTC neurons is determined by the solution of the following initial value problem Hasani et al. (2020):

$$\frac{d\boldsymbol{g}(t)}{dt} = -\left[\boldsymbol{l} + f(\boldsymbol{P}(t))\right] \cdot \boldsymbol{g}(t) + f(\boldsymbol{P}(t)) \cdot p_r, \tag{1}$$

where $t$ represents time, $\boldsymbol{l}$ denotes the leakage conductance vector of the LTC neurons Lechner et al. (2020), $\boldsymbol{P}(t)$ is the exogenous input signal, $f(\cdot)$ represents nonlinear functions of synaptic inputs, and $p_r$ indicates the reversal potential of LTC neurons.

To enhance the applicability of LTC neurons models in vision-related tasks, we reformulate Eq.1 into a computationally stable and learnable representation. By introducing two learnable parameters $\alpha_k = \boldsymbol{l} + f(\boldsymbol{P}_k)$ as the decay rate and $\beta_k = f(\boldsymbol{P}_k)$ as the modulation factor, we can derive an

expression for membrane potential evolution over time intervals:

$$\frac{d\boldsymbol{g}(t)}{dt} = -\alpha_k \cdot \boldsymbol{g}(t) + \beta_k \cdot p_r, \tag{2}$$

where $\alpha_k$ controls the rate of the signal attenuates, and $\beta_k$ modulates the efficiency of inputs transmission efficiency. Thus, the closed-form solution to this differential equation can be formulated as Hasani et al. (2022):

$$\boldsymbol{g}(t) \approx (\boldsymbol{g}_0 - p_r) e^{-\alpha_k t} \cdot \beta_k + p_r, \tag{3}$$

where $\boldsymbol{g}_0$ represents the initial membrane potential. Eq.1 and Eq.3 describe the basic behavior of LTC neurons, but present challenges for direct application to vision tasks. To be specific, Eq.2 assumes that the input $\boldsymbol{I}(t)$ remains piecewise constant over specific time intervals Hasani et al. (2022); Ayoub et al. (2024). This property is a reasonable assumption for causal signals that involve a sequential nature but poses challenges when transferring to non-causal data such as images Karn et al. (2024).

This MLP learns a nonlinear mapping that converts spatial distance relationships between patches into temporal intervals, effectively creating a manifold that preserves locality while enabling differential equation dynamics across the image structure. During training, the MLP develops a transformation that prioritizes quality-relevant spatial adjacencies as temporal proximities, allowing our continuous-time framework to process spatial relationships through biologically-inspired neural dynamics.

To address this, we flatten $\boldsymbol{M}$ through a multi-layer perceptron (MLP) layer to transform spatial positions into time parameters, which converts each window's position into a corresponding time step on a temporal axis. This MLP layer learns a nonlinear mapping that converts spatial distance relationships between patches into temporal intervals, creating a manifold that preserves locality while enabling differential equation dynamics across the image structure. During training, the MLP develops a transformation that prioritizes quality-relevant spatial adjacencies as temporal proximities, allowing our continuous-time framework to process spatial relationships through biologically-inspired neural dynamics. Next, partition the time parameters into $N$ sub-intervals, taking $k$-th time interval $[t_k, t_{k+1}]$ and the corresponding state $\boldsymbol{g}(t_k)$, Eq.2 can be parameterized as:

$$\tilde{\boldsymbol{g}}(t_{k+1}) \approx (\boldsymbol{g}(t_k) - p_r) e^{-\alpha_k \Delta t_k} \cdot \beta_k + p_r, \tag{4}$$

where $\Delta t_k = t_{k+1} - t_k$ represents the length of the interval. And to facilitate Eq.4 for stability of the tensor for large-scale image processing, we employ the (1,1)-order Padé approximant to $e^{-\alpha_k \Delta t_k}$ for exponential linearization:

$$e^{-\alpha_k \Delta t_k} \approx \frac{1 - \alpha_k \Delta t_k / 2}{1 + \alpha_k \Delta t_k / 2}. \tag{5}$$

By substituting Eq.5 into Eq.4 , we obtain a simplified closed-form solution:

$$\tilde{\boldsymbol{g}}(t_{k+1}) \approx (\boldsymbol{g}(t_k) - p_r) \left( \frac{1 - (\alpha_k \Delta t_k)/2}{1 + (\alpha_k \Delta t_k)/2} \right) \cdot \beta_k + p_r. \tag{6}$$

Subsequently, CFCTM updates the state $\boldsymbol{g}(t)$ of LTC neurons at each sub-intervals. For example, in $k$-th time interval $[t_k, t_{k+1}]$, the time-continuous reset gate $\boldsymbol{r}_k$ and update gate $\boldsymbol{u}_k$ are computed via the $k$-th neuronal input features $\boldsymbol{P}_k$ and the current state $\boldsymbol{g}(t_k)$ Hasani et al. (2022); Chahine et al. (2023). By applying Euler integration to Eq.2, we obtain information from the preceding $k-1$ temporal intervals and subsequently regulate the intermediate state $\tilde{\boldsymbol{g}}(t_k)$ within the $k$-th interval via $\boldsymbol{r}_k$. This mechanism enables $\boldsymbol{r}_k$ to selectively attenuate the information propagated from $\boldsymbol{g}(t_k)$ and previous temporal intervals. These operations can be formulated as in:

$$\tilde{\boldsymbol{g}}(t_k) = (1 - \boldsymbol{r}_k) \cdot \boldsymbol{g}(t_k) + \boldsymbol{r}_k \cdot \sum_{n=1}^{k-1} \frac{d\boldsymbol{g}(t_n)}{dt} \frac{\Delta t_n}{k}. \tag{7}$$

Next, to dynamically adapt the neuron's state transition based on the characteristics of the input visual features $\boldsymbol{P}_k$ at each time step, the $(k+1)$-th state of LTC neurons $\boldsymbol{g}(t_{k+1})$ can be calculated via $\tilde{\boldsymbol{g}}(t_{k+1})$, $\tilde{\boldsymbol{g}}(t_k)$, and $\boldsymbol{u}_k$:

$$\boldsymbol{g}(t_{k+1}) = (1 - \boldsymbol{u}_k) \cdot \tilde{\boldsymbol{g}}(t_k) + \boldsymbol{u}_k \cdot \tilde{\boldsymbol{g}}(t_{k+1}). \tag{8}$$

Finally, through the iterative application of Eq.8 across all time intervals, the CFCTM processes $\boldsymbol{P}_{\mathrm{w}}$ along with the corresponding time parameters and generates the dynamically modeled state features $\boldsymbol{G} \in \mathbb{R}^{YB \times T^2 \times L}$:

$$\boldsymbol{G} = \mathrm{CFCTM}(\boldsymbol{P}_{\mathrm{w}}, \mathrm{MLP}(\boldsymbol{M})), \tag{9}$$

where $\mathrm{MLP}(\cdot)$ indicates the multi-layer perceptron.

### 3.2.3 LIQUID SELF-ATTENTION

We then perform a series of dimensional transformations and information fusion on $\boldsymbol{G} \in \mathbb{R}^{NB \times T^2 \times L}$ obtained from CFCTM, including: (1) Adjust the channel number of $\boldsymbol{G}$ to $3L$ through a layer normalization operation. (2) Split the $\boldsymbol{G}$ into three groups of matrices along the channel dimension, including query $\boldsymbol{Q}$, key $\boldsymbol{K}$ and value $\boldsymbol{V}$ through the splitting operation, where $\boldsymbol{Q} = \{\boldsymbol{Q}_1, \ldots, \boldsymbol{Q}_h\}, \boldsymbol{K} = \{\boldsymbol{K}_1, \ldots, \boldsymbol{K}_h\}, \boldsymbol{V} = \{\boldsymbol{V}_1, \ldots, \boldsymbol{V}_h\} \in \mathbb{R}^{NB \times T^2 \times L}$. These operations are formally defined as in:

$$\boldsymbol{Q}, \boldsymbol{K}, \boldsymbol{V} = \mathrm{SP}\left(\mathrm{LN}\left(\boldsymbol{G}\right)\right), \tag{10}$$

where $\mathrm{SP}(\cdot)$ and $\mathrm{LN}(\cdot)$ indicate the splitting operation and layer normalization, respectively.

With these matrices, we perform the LSA operation following standard Transformer-based attention mechanisms Dosovitskiy et al. (2021); Liu et al. (2021), written by:

$$\mathrm{LSA}\left(\boldsymbol{Q}, \boldsymbol{K}, \boldsymbol{V}\right) = \mathrm{SoftMax}\left(\frac{\boldsymbol{Q}\boldsymbol{K}^{\mathrm{T}}}{\sqrt{L}} + \boldsymbol{M}\right)\boldsymbol{V}. \tag{11}$$

The LSA output is combined with the original patch embeddings through a residual connection, followed by another MLP and normalization layer to produce the final output of the LTCFormer block:

$$\widetilde{\boldsymbol{P}} = \mathrm{LSA}\left(\boldsymbol{Q}, \boldsymbol{K}, \boldsymbol{V}\right) + \boldsymbol{P}_{\mathrm{Ed}}, \tag{12}$$

$$\boldsymbol{P}_{\mathrm{o}} = \mathrm{MLP}(\mathrm{LN}(\widetilde{\boldsymbol{P}})) + \widetilde{\boldsymbol{P}}. \tag{13}$$

### 3.3 LIQUID IMAGE QUALITY-AWARE FRAMEWORK

We subsequently adapt the LTCFormer for BIQA tasks by employing the LTCFormer framework for hierarchical feature extraction and a Multi-Scale Image Quality-Aware Decoder (MIQAD) for quality prediction. As illustrated in Figure 2, the patch tokens are processed through four LTCFormer stages, denoted as $S_1$, $S_2$, $S_3$, and $S_4$. The framework yields feature maps $\boldsymbol{F}_i \in \mathbb{R}^{B \times (H/2^{i+1}) \times (W/2^{i+1}) \times C_i}$ at the $i$-th stage, where $C_i = 2^{i-1} \times C_1$ is the feature dimension at the $i$-th stage. The extracted feature maps $\boldsymbol{F}_1$, $\boldsymbol{F}_2$, $\boldsymbol{F}_3$, and $\boldsymbol{F}_4$ from these four stages are then fed into MIQAD, which characterizes image quality from multi-perspectives.

Within MIQAD, we implement a multi-scale quality regression approach to effectively leverage the hierarchical features extracted by the LTCFormer framework. This mechanism is inspired by the multi-faceted nature of human visual perception, where quality assessment occurs simultaneously across different perceptual dimensions. In human visual evaluation, observers naturally assess images at multiple scales from fine-grained details to overall compositional harmony, and different observers often prioritizing different aspects of visual quality. MIQAD's multi-component quality assessment structure methodically reproduces this cognitive process, as shown in Figure 2, each feature map $\boldsymbol{F}_i$, $i \in \{1, 2, 3, 4\}$ undergoes a dedicated quality regression pathway to generate scale-specific quality scores $S_i$. Specifically, each feature map $\boldsymbol{F}_i$ is processed through a scale-specific quality regression module consisting of global average pooling followed by a MLP. This process can be formulated as:

$$S_i = \mathrm{MLP}_i(\mathrm{GAP}(\boldsymbol{F}_i)), \quad i \in \{1, 2, 3, 4\}, \tag{14}$$

where GAP represents global average pooling operation and $\mathrm{MLP}_i$ denotes the quality regression network for the $i$-th scale. With the score from coarse to fine, MIQAD can achieve a comprehensive evaluation of the image quality, thus reducing the prediction uncertainty. The final image quality score $\mathcal{S}$ is obtained by averaging these four scale-specific quality scores from Eq.14:

$$\mathcal{S} = \frac{1}{4}\sum_{i=1}^{4} S_i. \tag{15}$$

Table 1: Quantitative comparison based on average SRCC and PLCC. Bold values denote the best result per dataset. We have detailed the trainable parameter values of mainstream SOTA models.

| Method | Train Param. | LIVE | | CSIQ | | TID2013 | | LIVEC | | KonIQ | | LIVEFB | |
|---|---|---|---|---|---|---|---|---|---|---|---|---|---|
| | | PLCC | SRCC | PLCC | SRCC | PLCC | SRCC | PLCC | SRCC | PLCC | SRCC | PLCC | SRCC |
| BRISQUE | - | 0.944 | 0.929 | 0.748 | 0.812 | 0.571 | 0.626 | 0.629 | 0.629 | 0.685 | 0.681 | 0.341 | 0.303 |
| ILNIQE | - | 0.906 | 0.902 | 0.865 | 0.822 | 0.648 | 0.521 | 0.508 | 0.508 | 0.537 | 0.523 | 0.332 | 0.294 |
| BIECON | - | 0.961 | 0.958 | 0.823 | 0.815 | 0.762 | 0.717 | 0.613 | 0.613 | 0.654 | 0.651 | 0.428 | 0.407 |
| MEON | - | 0.955 | 0.951 | 0.864 | 0.852 | 0.824 | 0.808 | 0.710 | 0.697 | 0.628 | 0.611 | 0.394 | 0.365 |
| DBCNN | - | 0.971 | 0.968 | 0.959 | 0.946 | 0.865 | 0.816 | 0.869 | 0.851 | 0.884 | 0.875 | 0.551 | 0.545 |
| MetaIQA | - | 0.959 | 0.960 | 0.908 | 0.899 | 0.868 | 0.856 | 0.802 | 0.835 | 0.856 | 0.887 | 0.507 | 0.54 |
| P2P-BM | - | 0.958 | 0.959 | 0.902 | 0.899 | 0.856 | 0.862 | 0.842 | 0.844 | 0.885 | 0.872 | 0.598 | 0.526 |
| HyperIQA | 27M | 0.966 | 0.962 | 0.942 | 0.923 | 0.858 | 0.840 | 0.882 | 0.859 | 0.917 | 0.906 | 0.602 | 0.544 |
| MUSIQ | 27M | 0.911 | 0.940 | 0.893 | 0.871 | 0.815 | 0.773 | 0.828 | 0.785 | 0.928 | 0.916 | 0.661 | 0.566 |
| TReS | 152M | 0.968 | 0.969 | 0.942 | 0.922 | 0.883 | 0.863 | 0.882 | 0.859 | 0.928 | 0.915 | 0.625 | 0.554 |
| CLIP-IQA+ | - | - | - | - | - | - | - | 0.832 | 0.805 | 0.909 | 0.895 | 0.593 | 0.575 |
| Q-Align | 8.2B | - | - | 0.936 | 0.915 | - | - | 0.921 | **0.931** | 0.934 | 0.935 | - | - |
| Re-IQA | 48M | 0.971 | 0.970 | 0.96 | 0.947 | 0.861 | 0.804 | 0.854 | 0.840 | 0.923 | 0.914 | - | - |
| DEIQT | 24M | 0.982 | 0.980 | 0.963 | 0.946 | 0.908 | 0.892 | 0.894 | 0.875 | 0.934 | 0.921 | 0.663 | 0.571 |
| QFM-IQM | 24M | 0.983 | 0.981 | 0.965 | 0.954 | - | - | 0.913 | 0.891 | 0.936 | 0.922 | 0.667 | 0.567 |
| LoDa | 9M | 0.979 | 0.975 | - | - | 0.901 | 0.869 | 0.899 | 0.876 | 0.944 | 0.932 | 0.679 | 0.578 |
| Align-IQA | 35M | 0.987 | 0.985 | 0.981 | 0.975 | 0.960 | 0.955 | 0.916 | 0.905 | 0.932 | 0.923 | - | - |
| LQMamba-B | 94M | 0.959 | 0.951 | 0.915 | 0.889 | 0.965 | 0.964 | 0.913 | 0.888 | **0.947** | **0.933** | 0.675 | 0.582 |
| **LIQT (Ours)** | **7M** | **0.988** | **0.985** | **0.983** | **0.976** | **0.964** | **0.958** | **0.925** | 0.919 | 0.939 | 0.926 | **0.682** | **0.586** |

## 3.4 Loss Function

We optimize our architecture by minimize the $L_1$ loss for BIQA, which can be formulated as:

$$\mathcal{L} = \|s_p - s_{gt}\|_1, \tag{16}$$

where $\|\cdot\|_1$ denotes the $L_1$ norm, $s_p$ denotes the predicted quality score, and $s_{gt}$ represents the corresponding ground truth score.

## 4 Experiments

### 4.1 Experimental Settings

**Implementation Details.** For LIQT training, we followed the typical training strategy outlined in DEIQT Qin et al. (2023) to randomly standardize an input image with a pixel resolution of $224 \times 224$. Our model uses LTCFormer with MIQAD decoder to obtain MOS scores. Training runs for 9 epochs with $2 \times 10^{-4}$ learning rate, applying tenfold decay every 3 epochs. Batch size varies with dataset size. For each dataset, 80% images were used for training and the remaining 20% images were utilized for testing. We repeated this process 10 times to mitigate the performance bias and the medians of SRCC and PLCC were reported, following Qin et al. (2023). All experiments run on four NVIDIA 4090 GPUs.

**Compared Methods.** We compared 18 popular or state-of-the-art (SOTA) methods, including CNN-based approaches such as BRISQUE Mittal et al. (2012), ILNIQE Zhang et al. (2015), BIECON Kim & Lee (2016), DBCNN Zhang et al. (2018), MetaIQA Zhu et al. (2020), P2P-BM Ying et al. (2020), HyperIQA Su et al. (2020), and Re-IQA Saha et al. (2023). Our comparison also covers Transformer-based methods like MUSIQ Ke et al. (2021), TReS Golestaneh et al. (2022), DEIQT Qin et al. (2023), and QFM-IQM Li et al. (2025), as well as CLIP-based methods such as CLIP-IQA+. Additionally, we evaluated hybrid CNN and ViT approaches including TReS Golestaneh et al. (2022), LoDa Xu et al. (2024), Align-IQA Yang et al. (2024) along with the LLM-based Q-Align Wu et al. (2023), and LQMamba-B Guan et al. (2025). We report detailed SRCC and PLCC performance across multiple datasets, with results sourced either from original papers or reproduced using publicly available code.

**Benchmark Datasets.** We evaluated the LIQT on six public Image Quality Assessment (IQA) datasets. Among these, LIVEC Ghadiyaram & Bovik (2016) and KonIQ-10k Hosu et al. (2020)

contain authentic distortions, while LIVE Sheikh et al. (2006), CSIQ Chandler (2010), TID2013 Ponomarenko et al. (2015), and LIVEFB Ying et al. (2020) feature synthetic distortions. The LIVEC dataset comprises 1,162 images with diverse real-world distortions, whereas KonIQ-10k includes 10,073 images sourced from open multimedia repositories. LIVEFB represents the largest real distortion dataset, containing 39,810 images. Synthetic datasets typically contain a limited number of pristine images with applied artificial distortions such as Gaussian blur and JPEG compression. The LIVE and CSIQ datasets include 779 and 866 synthetic images, respectively, covering 5 and 6 distortion categories. TID2013 offers more extensive collections, with 3,000 images exhibiting 24 distortion types.

**Evaluation Metrics.** For evaluation metrics, we employed the Spearman Rank Correlation Coefficient (SRCC) and Pearson Linear Correlation Coefficient (PLCC) to assess monotonicity and accuracy, with values ranging from -1 to 1, where coefficients approaching 1 indicate superior predictive performance.

## 4.2 QUANTITATIVE AND QUALITATIVE COMPARISON

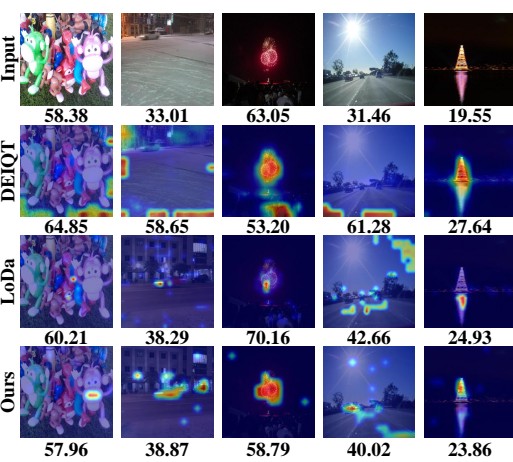

Figure 3: Activation maps of DEIQT, LoDa, and LIQT, drawn using the Grad-CAM method Selvaraju et al. (2017). The figure also shows the MOS. Our LIQT model is designed to focus more on spatiotemporal feature aggregation of images, thereby improving image quality prediction performance. Rows 1-4 in the figure represent the input images, DEIQT, LoDa, and LIQT's CAMs, respectively.

In Section 4.1, we reported on competing models and their corresponding backbones, presenting detailed comparison results in Table 1. LIQT achieved highly competitive performance among existing models. Since these datasets cover a wide range of distortion types and image content, achieving competitive performance across them is challenging. Notably, we achieved the best results on the most challenging LIVEFB dataset, as the prevalent local distortions in LIVEFB align with the spatiotemporal aggregation characteristics of liquid neural networks. Meanwhile, it should be emphasized that we only used 7M trainable parameters, as sequential tasks inherently possess memory capabilities and do not require excessive training parameters. This better matches the characteristics of image quality assessment, that estimators can remember most image features without needing multiple comparisons during evaluation.

In addition to the quantitative comparison, we present qualitative comparison results in Figure 3 to depict the visual results of the activation maps from DEIQT, LoDa, and LIQT. DEIQT's activation maps primarily focus on high-contrast regions and object boundaries, LoDa shows improved attention distribution. LIQT's activation maps demonstrate more comprehensive coverage of perceptually important regions, attending to both structural elements and texture details. This enhanced spatiotemporal feature aggregation aligns better with human visual perception, as confirmed by the Mean Opinion Scores (MOS).

## 4.3 CROSS-DATASET EVALUATION

To further evaluate the generalization ability of LIQT, we conducted cross-dataset validation. Specifically, our model was trained on one dataset and then tested on another dataset without any fine-tuning or parameter adaptation. To ensure simplicity and universality, we conducted several sets of experiments. The experimental results are represented by the average SRCC values on these datasets. Encouragingly, LIQT achieved state-of-the-art performance in all experiments, despite having fewer trainable parameters than the compared methods. Through spatiotemporal aggregation of streaming neural networks, our model can better understand key quality representation spaces

Table 2: SRCC on the cross datasets validation. The best performances are highlighted in boldface.

| Training | LIVEFB | | LIVEC | KonIQ | LIVE | CSIQ |
|---|---|---|---|---|---|---|
| Testing | KonIQ | LIVEC | KonIQ | LIVEC | CSIQ | LIVE |
| DBCNN | 0.716 | 0.724 | **0.754** | 0.755 | 0.758 | 0.877 |
| P2P-BM | 0.755 | 0.738 | 0.740 | 0.770 | 0.712 | - |
| TReS | 0.713 | 0.740 | 0.733 | 0.786 | 0.761 | - |
| DEIQT | 0.733 | 0.781 | 0.744 | 0.794 | 0.781 | 0.932 |
| LoDa | 0.763 | 0.805 | 0.745 | **0.811** | - | - |
| **LIQT (Ours)** | **0.771** | **0.810** | 0.741 | 0.807 | **0.792** | **0.937** |

Table 3: Ablation study on LIVEC and KonIQ datasets. Each row shows the performance with different combinations of components: $\alpha_k$, $\beta_k$, CFCTM, LSA, and MIQAD. The best performances are highlighted in boldface.

| $\alpha_k$ | $\beta_k$ | **CFCTM** | **LSA** | **MIQAD** | **LIVEC** | | **KonIQ** | |
|---|---|---|---|---|---|---|---|---|
| | | | | | PLCC | SRCC | PLCC | SRCC |
| | | | | | 0.818 | 0.806 | 0.842 | 0.833 |
| | | | | ✓ | 0.832 | 0.824 | 0.857 | 0.849 |
| ✓ | | | ✓ | ✓ | 0.853 | 0.849 | 0.872 | 0.869 |
| | ✓ | | ✓ | ✓ | 0.844 | 0.835 | 0.863 | 0.856 |
| | | ✓ | ✓ | ✓ | 0.867 | 0.854 | 0.877 | 0.868 |
| ✓ | ✓ | | ✓ | ✓ | 0.876 | 0.870 | 0.881 | 0.874 |
| ✓ | | ✓ | ✓ | ✓ | 0.903 | 0.894 | 0.922 | 0.913 |
| | ✓ | ✓ | ✓ | ✓ | 0.899 | 0.887 | 0.905 | 0.896 |
| ✓ | ✓ | ✓ | | ✓ | 0.880 | 0.874 | 0.885 | 0.878 |
| ✓ | ✓ | ✓ | ✓ | ✓ | **0.925** | **0.919** | **0.939** | **0.926** |

rather than the decoupled collapse of existing neural networks, thus achieving robust generalization ability with a streaming structure.

## 4.4 ABLATION STUDY

Table 3 presents a comprehensive ablation analysis of LIQT's key components on LIVEC and KonIQ datasets, examining individual and combined contributions of decay rate $\alpha_k$, modulation factor $\beta_k$, CFCTM, LSA, and MIQAD modules. The first row represents a pure Swin Transformer baseline without any of our proposed components. The results demonstrate the progressive contribution of each component to the overall performance. The pure Swin Transformer baseline serves as our reference point, with MIQAD alone providing improvements, demonstrating the effectiveness of multi-scale quality assessment. CFCTM provides the most significant individual contribution among liquid neural components, and removing LSA while maintaining other components substantially reduces performance, highlighting the critical role of LSA.

## 5 CONCLUSION

In this paper, we introduced LIQT, a novel BIQA framework that addresses the STRD challenge by integrating biomimetic neural mechanisms. We developed CFCTM that reformulates liquid neuronal dynamics into stable closed-form solutions, enabling continuous-time processing aligned with human visual perception. LTCFormer dynamically modulates feature extraction based on local image features through LSA mechanism. MIQAD effectively emulates the multi-faceted nature of human visual assessment through scale-specific quality regression pathways. Experimental results on six benchmark datasets spanning various distortion types and image content demonstrate that LIQT consistently outperforms state-of-the-art BIQA methods.

ETHICS STATEMENT

This work adheres to the ICLR Code of Ethics. In this study, no human subjects or animal experimentation was involved. All datasets used were sourced in compliance with relevant usage guidelines, ensuring no violation of privacy. We have taken care to avoid any biases or discriminatory outcomes in our research process. No personally identifiable information was used, and no experiments were conducted that could raise privacy or security concerns. We are committed to maintaining transparency and integrity throughout the research process.

REPRODUCIBILITY STATEMENT

To ensure the reproducibility of our research, the source code for the proposed LIQT model, along with training and evaluation scripts, is provided in the Supplementary Material. The implementation details, hyper-parameters, and experimental setup described in Section 4.1 of the main paper are sufficient to replicate the reported results. Additionally, the six IQA benchmark datasets are publicly available, ensuring consistent and reproducible evaluation results. We believe these measures will enable other researchers to reproduce our work and further advance the field.

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

## A APPENDIX

### A.1 THE USE OF LARGE LANGUAGE MODELS (LLMS)

Large Language Models (LLMs) were used to aid in the writing and polishing of the manuscript. Specifically, we used an LLM to assist in refining the language, improving readability, and ensuring clarity in various sections of the paper. The model helped with tasks such as sentence rephrasing, grammar checking, and enhancing the overall flow of the text.

It is important to note that the LLM was not involved in the ideation, research methodology, or experimental design. All research concepts, ideas, and analyses were developed and conducted by the authors. The contributions of the LLM were solely focused on improving the linguistic quality of the paper, with no involvement in the scientific content or data analysis.

The authors take full responsibility for the content of the manuscript, including any text generated or polished by the LLM. We have ensured that the LLM-generated text adheres to ethical guidelines and does not contribute to plagiarism or scientific misconduct.

### A.2 DERIVATION OF CLOSED-FORM SOLUTIONS FOR LIQUID TIME-CONSTANT NEURONS

The fundamental dynamics of LTC neurons are governed by the differential equation presented in Eq.1 Hasani et al. (2022). To derive the closed-form solution that enables efficient computation, we begin with the differential equation governing the membrane potential $\boldsymbol{g}(t)$ of an LTC neuron:

$$\frac{d\boldsymbol{g}(t)}{dt} = -\left[\boldsymbol{l} + f\left(\boldsymbol{P}(t)\right)\right] \cdot \boldsymbol{g}(t) + f\left(\boldsymbol{P}(t)\right) \cdot p_r, \tag{17}$$

where $\boldsymbol{l}$ is the leakage conductance vector of the LTC neurons Lechner et al. (2020), $f(\boldsymbol{P}(t))$ is a nonlinear function of the exogenous input signal $\boldsymbol{P}(t)$, and $p_r$ is the reversal potential of LTC neurons. For computational tractability, we assume that over a small time interval $[t_k, t_{k+1}]$, the input signal $\boldsymbol{P}(t)$ is piecewise constant Lechner et al. (2020). This allows us to define two learnable, input-dependent parameters for that interval:

- The decay rate: $\alpha_k = \boldsymbol{l} + f(\boldsymbol{P}_k)$.
- The modulation factor: $\beta_k = f(\boldsymbol{P}_k)$.

Substituting these into Eq.17 yields a simplified linear ordinary differential equation (ODE):

$$\frac{d\boldsymbol{g}(t)}{dt} = -\alpha_k \cdot \boldsymbol{g}(t) + \beta_k \cdot p_r. \tag{18}$$

To solve this, we first rearrange it into the standard form for a first-order linear ODE:

$$\frac{d\boldsymbol{g}(t)}{dt} + \alpha_k \boldsymbol{g}(t) = \beta_k p_r. \tag{19}$$

The general solution is the sum of the homogeneous solution $\boldsymbol{g}_h(t)$ and a particular solution $\boldsymbol{g}_p(t)$. The homogeneous part of the equation is:

$$\frac{d\boldsymbol{g}_h}{dt} + \alpha_k \boldsymbol{g}_h = 0. \tag{20}$$

The solution to this separable equation is $\boldsymbol{g}_h(t) = Ce^{-\alpha_k t}$, where $C$ is the constant of integration. For the particular solution, since the right-hand side of Eq.19 is a constant, we assume a constant particular solution $\boldsymbol{g}_p(t) = Z$. Substituting this into Eq.19 gives:

$$\frac{dZ}{dt} + \alpha_k Z = \beta_k p_r. \tag{21}$$

Since $Z$ is a constant, its derivative is zero, which simplifies to $\alpha_k Z = \beta_k p_r$, so $Z = \frac{\beta_k p_r}{\alpha_k}$. Therefore, the general solution is:

$$\boldsymbol{g}(t) = \boldsymbol{g}_h(t) + \boldsymbol{g}_p(t) = Ce^{-\alpha_k t} + \frac{\beta_k p_r}{\alpha_k}. \tag{22}$$

We determine the integration constant $C$ using the initial condition at the start of the interval, $t = 0$, where the membrane potential is $\boldsymbol{g}(0) = \boldsymbol{g}_0$. Substituting this into the general solution gives:

$$\boldsymbol{g}_0 = Ce^0 + \frac{\beta_k p_r}{\alpha_k} \implies C = \boldsymbol{g}_0 - \frac{\beta_k p_r}{\alpha_k}. \tag{23}$$

Substituting this expression for $C$ back into the general solution (Eq.22) yields the exact solution for the interval:

$$\boldsymbol{g}(t) = \left(\boldsymbol{g}_0 - \frac{\beta_k p_r}{\alpha_k}\right)e^{-\alpha_k t} + \frac{\beta_k p_r}{\alpha_k}. \tag{24}$$

This solution can be rearranged by expanding the terms and factoring out $\boldsymbol{g}_0$ and $\frac{\beta_k p_r}{\alpha_k}$:

$$\boldsymbol{g}(t) = \boldsymbol{g}_0 e^{-\alpha_k t} + \frac{\beta_k p_r}{\alpha_k}(1 - e^{-\alpha_k t}). \tag{25}$$

While Eq. 25 represents the exact solution, the term $(\beta_k p_r)/\alpha_k$ can introduce numerical instability if $\alpha_k = \boldsymbol{l} + f(\boldsymbol{P}_k)$ becomes close to zero during training. This can happen if the leakage $\boldsymbol{l}$ is small and the input activation $f(\boldsymbol{P}_k)$ is also close to zero, leading to potential division-by-zero errors and training instability. To ensure a robust implementation, we adopt an alternative, more numerically stable formulation inspired by prior work Lechner et al. (2020); Hasani et al. (2022); Karn et al. (2024). This formulation approximates the steady-state value $\frac{\beta_k p_r}{\alpha_k}$ with an interpolation controlled by $\beta_k$, leading to the following equivalent but more stable parameterization:

$$\boldsymbol{g}(t) \approx (\boldsymbol{g}_0 - p_r)e^{-\alpha_k t} \cdot \beta_k + p_r. \tag{26}$$

This form represents the state at time $t$ as an interpolation between the initial state $\boldsymbol{g}_0$ and the reversal potential $p_r$, driven by input-dependent dynamics. It avoids explicit division by $\alpha_k$, replacing it with multiplications that are numerically more robust. This closed-form solution allows us to bypass computationally expensive numerical integration methods like Runge-Kutta, while preserving the core dynamics of the continuous-time system Hasani et al. (2022); Karn et al. (2024).

### A.3 ANALYSIS OF THE PADÉ APPROXIMANT IN CFCTM

In our Closed-Form Continuous-Time Module (CFCTM), we approximate the exponential term $e^{-\alpha_k \Delta t_k}$ using the (1,1)-order Padé approximant, as shown in Eq. 5. The approximant is given by:

$$e^{-x} \approx \frac{1 - x/2}{1 + x/2}. \tag{27}$$

We select this method over alternatives such as the first-order Taylor series expansion ($e^{-x} \approx 1 - x$) based on the following considerations.

Regarding numerical stability, the Taylor expansion becomes negative when $x > 1$, which may lead to unstable liquid neural dynamics when the decay term $\alpha_k \Delta t_k$ is large. The Padé approximant remains positive for all $x \geq 0$ and asymptotically approaches -1 as $x \to \infty$. Its value is therefore bounded, which helps prevent unstable state dynamics during training.

In terms of approximation accuracy, the Padé approximant matches the Taylor series of $e^{-x}$ up to the second order, providing higher approximation accuracy over a wider range of $x$ values compared to the first-order Taylor expansion. This allows the discretized dynamics of our model to more closely approximate the underlying continuous-time system.

From a computational efficiency perspective, this approximation requires only basic arithmetic operations (division, addition, and subtraction), which are already optimized on modern hardware such as GPUs. This enables our CFCTM to avoid the computational overhead of repeatedly calculating the exponential function during each forward pass.

