# OpenReview forum: "LIQT: Bridging Liquid Neural Dynamics and Human Perceptual Mechanisms for Blind Image Quality Assessment"
_ICLR.cc/2026/Conference — ICLR 2026 Conference Withdrawn Submission_

### Official Review · Reviewer_GrXp · 2025-10-23

**Soundness:** 3
**Presentation:** 3
**Contribution:** 2
**Rating:** 4
**Confidence:** 4

**Summary:**

This paper proposes the Liquid Image Quality Transformer (LIQT), a framework for BIQA that mimics the human visual system. LIQT integrates Liquid Neural Networks (LNNs) into a Transformer architecture to better model the continuous-time neural dynamics of human perception. The framework uses a Liquid Time-Constant Transformer (LTCFormer) with a novel Closed-Form Continuous-Time Module (CFCTM) to adaptively process image features based on their local characteristics. Additionally, a Multi-Scale Image Quality-Aware Decoder (MIQAD) is used to aggregate features at different scales for a more comprehensive quality prediction. Experimental results show that this lightweight model outperforms state-of-the-art methods on six benchmark BIQA datasets.

**Strengths:**

1. The manuscript is well-written, with a clear logical flow and a well-organized structure.

2. The primary strength lies in the novelty of the LIQT framework. It’s a lightweight IQA model and demonstrates good performance.

**Weaknesses:**

1. The central weakness lies in the paper's core motivation. The authors introduce the "Spatio-Temporal Representation Disentanglement (STRD)" challenge to justify the use of LNNs. However, the definition and applicability of this "spatio-temporal" problem to the static, non-sequential domain of image quality assessment are insufficiently justified and questionable.

2. The paper fails to provide a clear and compelling justification for its method of transforming a spatial problem into a temporal one. The model processes image patches sequentially, but the mapping from spatial patch location to a "temporal" order is not adequately explained or defended. It is unclear how this specific sequential processing is analogous to HVS dynamics for a static image.

3. The cross-dataset evaluation in Table 2 could be strengthened. Many recent VLM-based IQA models (e.g., DeQA-Score, Q-Insight) demonstrate robust generalization by training only on the KonIQ-10k dataset and testing on all others. Including this specific experimental setting would provide a more direct and rigorous comparison of LIQT's generalization capabilities against contemporary methods.

4. The benchmark comparison in Tables 1 and 2 should include results on newer and more challenging datasets, such as PIPAL and AGIQA.

**Questions:**

If the "historical data" and their temporal information are essential, as the LNN-based motivation implies, the model's output should be sensitive to the patch processing order.
How sensitive is the model's performance to the patch processing order? For instance, have the authors conducted an ablation study comparing the standard sequential order to a randomized patch order?

---

### Official Review · Reviewer_vvcc · 2025-10-30

**Soundness:** 3
**Presentation:** 2
**Contribution:** 2
**Rating:** 2
**Confidence:** 4

**Summary:**

The paper addresses Blind Image Quality Assessment (BIQA), where no reference image is available and the model must infer perceptual quality solely from the degraded image. The authors propose Liquid Image Quality Transformer (LIQT), which integrates Liquid Self-Attention (LSA) and Multi-Scale Image Quality-Aware Decoder (MIQAD).

**Strengths:**

1. An IQA model using liquid attention is proposed.
2. The proposed method achieves state-of-the-art performance.

**Weaknesses:**

1. The motivation is weak. As shown in Fig.1, it is hard to connect liquid attention closer to HVS. Why CNN and transformer do not have memory. They have memorized the historical data in their parameters. Can LNN memorize new data without training?
2. The generalization ability is limited to some extent.

**Questions:**

1. Can LNN memorize new data without training?
2. What is “train param.”. Is it same with the “param.”? Can you report MAC/GFlops comparison or test time comparison with other models?

---

### Official Review · Reviewer_CkYg · 2025-10-31

**Soundness:** 2
**Presentation:** 2
**Contribution:** 2
**Rating:** 4
**Confidence:** 5

**Summary:**

This manuscript proposes LIQT, a blind image quality assessment (BIQA) model that blends Liquid Neural Network (LNN) dynamics with a Transformer backbone. The core technical pieces are a Closed Form Continuous Time Module (CFCTM)—intended to reparameterize liquid time constant (LTC) neurons via a closed form with a Padé (1,1) approximation. Results on six IQA datasets show strong correlations.

**Strengths:**

1. Conceptual novelty: A clear, biomimetic rationale for using continuous‑time dynamics on spatial tokens.

2. Compact, competitive model: 7 M parameters with strong PLCC/SRCC on several datasets

**Weaknesses:**

1. While the introduction emphasizes modeling the human visual system’s ability to integrate historical perceptual experiences through spatio-temporal memory consolidation—suggesting an intention to capture long-term or cross-image temporal dependencies (see Fig. 1(a))—the actual implementation of LIQT does not realize this form of memory. In the methods section, the so-called “temporal” dynamics are created by converting spatial adjacency within each image window into a pseudo-temporal sequence, which is then processed by the Closed-Form Continuous-Time Module (CFCTM). The model resets its state for every new image and maintains no persistent or cross-sample memory. Consequently, the claimed alignment with human temporal memory or experience integration is conceptually overstated. The method effectively performs intra-image sequential integration rather than inter-image temporal learning, weakening the coherence between the biological motivation and the engineered mechanism.

2. Equations (7)–(8) introduce time-continuous reset and update gates but do not explain how the gating variables are obtained.

3. The MLP responsible for mapping relative positions to temporal parameters plays a key role in addressing the STRD problem. However, the paper does not clarify important implementation details—such as whether the generated temporal sequence is monotonic or normalized, how the number of temporal
segments is determined, or whether the total duration of these segments is constrained.

4. The paper states LIQT “outperforms … across all datasets,” but Table 1 shows stronger results for LQMamba‑B on KonIQ‑10k and TID2013.

5. The paper emphasizes parameter count but does not report FLOPs or training time vs. baselines.

**Questions:**

1. The qualitative analysis requires deeper interpretation and clearer justification. The current evidence does not sufficiently substantiate the claim that the LIQT model effectively captures or emphasizes spatiotemporal feature aggregation within images

2. The overall presentation of the paper lacks clarity and precision. The writing is occasionally verbose and conceptually inconsistent, which obscures the main ideas and technical contributions. Several figures are not professionally formatted or adequately annotated, limiting their explanatory value. Furthermore, the conclusions and claimed contributions are not clearly delineated from the methodological description and tend to be overstated relative to the actual empirical evidence provided.

---

### Official Review · Reviewer_wFos · 2025-10-31

**Soundness:** 2
**Presentation:** 2
**Contribution:** 2
**Rating:** 4
**Confidence:** 4

**Summary:**

The paper introduces LIQT, a novel Blind Image Quality Assessment (BIQA) framework that bridges liquid neural dynamics and human perceptual mechanisms. LIQT integrates Liquid Neural Networks (LNNs) with Transformer-based architectures to capture the adaptive spatio-temporal integration of degradation patterns in the human visual system. The framework incorporates Liquid Self-Attention (LSA) equipped with a Closed-Form Continuous-Time Module (CFCTM) to dynamically modulate feature extraction based on local image features. A Multi-Scale Image Quality-Aware Decoder (MIQAD) aggregates multi-scale features for comprehensive quality regression. The paper demonstrates the superior performance of LIQT over state-of-the-art methods on six benchmark datasets.

**Strengths:**

1) The LIQT framework is well-designed and technically sound. The CFCTM and MIQAD components are innovative and effective.
2) The experimental results demonstrate the superior performance of LIQT over state-of-the-art methods.
3) The paper is generally well-written and easy to follow.

**Weaknesses:**

1) While the paper mentions the advantages of LNNs, it could more explicitly articulate why they are particularly well-suited for BIQA compared to other approaches.
2) The technical details of the CFCTM and LSA mechanisms could be more accessible to a broader audience. Provide more intuitive explanations and visualizations.
3) The ablation studies in Table 3 are helpful, but they could be more comprehensive. Consider ablating different components of the CFCTM and MIQAD modules.
4) While the paper claims computational efficiency, a more detailed analysis of the computational complexity of LIQT compared to other methods would be beneficial.
5) The motivation for the window tokenization approach could be clearer. Why is this necessary for modeling spatio-temporal relationships?

**Questions:**

6) Expand the cross-dataset evaluation in Table 2 to include more comprehensive testing scenarios. Specifically, evaluate the performance of models trained on authentic distortion datasets (e.g., LIVEC, KonIQ-10k, LIVEFB) and tested on synthetic distortion datasets (e.g., LIVE, CSIQ, TID2013), and vice versa. This will provide valuable insights into the generalizability of LIQT across different types of distortions.

Expand the cross-dataset evaluation in Table 2 to include more comprehensive testing scenarios. Specifically, evaluate the performance of models trained on authentic distortion datasets (e.g., LIVEC, KonIQ-10k, LIVEFB) and tested on synthetic distortion datasets (e.g., LIVE, CSIQ, TID2013), and vice versa. This will provide valuable insights into the generalizability of LIQT across different types of distortions.

**Details Of Ethics Concerns:**

N.A.

---

### Note · Authors · 2025-11-12

**Comment:**

We believe that there are some areas in the manuscript that require further improvement, we have decided to withdraw our paper after careful deliberation and discussion. We wish to express our appreciation to the reviewers, ACs, and SACs for their valuable time and insightful comments on this submission.

**Withdrawal Confirmation:**

I have read and agree with the venue's withdrawal policy on behalf of myself and my co-authors.